# Effects of the dialysate calcium concentrations and mineral bone disease treatments on mortality in The French Renal Epidemiology and Information Network (REIN) registry

Oriane Lambert[1], Cécile Couchoud[2], Marie Metzger[1], Gabriel Choukroun[3], Christian Jacquelinet[1], Lucile Mercadal[1,4]*

1 CESP, Centre for Research in Epidemiology and Population Health, Univ Paris-Saclay, Univ Paris Sud, UVSQ, INSERM UMRS, Villejuif, France, 2 Agence de Biomédecine, Saint Denis, France, 3 Nephrology, Dialysis & Transplantation Department, CHU Amiens, INSERM UMR, Jules Verne University of Picardie, Amiens, France, 4 Nephrology Department, Pitié-Salpêtrière Hospital, AP-HP, Paris, France

* lucile.mercadal@aphp.fr

**Data Availability Statement:** All data used for this research were extracted from the REIN registry,

## Abstract

### Background

In patients on hemodialysis (HD), the various chemical elements in the dialysate may influence survival rates. In particular, calcium modifies mineral and bone metabolism and the vascular calcification rate. We studied the influence of the dialysate calcium concentration and the treatments prescribed for mineral bone disease (MBD) on survival.

### Methods

All patients in REIN having initiated HD from 2010 to 2013 were classified according to their exposure to the different dialysate calcium concentrations in their dialysis unit. Data on the individual patients' treatments for MBD were extracted from the French national health database. Cox proportional hazard models were used to estimate mortality hazard ratios (HR) associated with time-dependent exposure to dialysate calcium concentrations and MBD therapies, adjusted for comorbidities, laboratory and technical data.

### Results

Dialysate calcium concentration of 1.5 mmol/L was used by 81% of the dialysis centers in 2010 and in 83% in 2014. Most centers were using several formulas in up to 78% for 3 formulas in 2010 to 86% in 2014. In full adjusted Cox survival analyses, the percentage of calcium >1.5 mmol/L and <1.5 mmol/l by center and the number of formula used per center were not associated with survival. Depending on the daily dose used, the MBD therapies were associated with survival improvement for calcium, native vitamin D, active vitamin D, sevelamer, lanthanum and cinacalcet in the second and third tertiles of dose.

coordinated and supported by the French
Biomedecine Agency. The access to national data
is regulated by a scientific committee of French
Biomedecine Agency which analyzes each request,
and so cannot be made publicly available due to
legal restrictions. Data are available upon request.
If readers need information about the data from the
REIN registry, they can contact Dr. Cecile
Couchoud and Soraya SEKOURI who coordinate
the REIN at the national level (email address: cecile.
couchoud@biomedecine.fr, soraya.
sekouri@biomedecine.fr).

**Funding:** The study was funded by the Agence de
la Biomédecine in 2016 and by the Société
Francophone Néphrologie Dialyse et
Transplantation in 2017 (Baxter grant). Author LM
received fees for a medical speech about the
results of the study at an AMGEN conference. The
funders had no role in study design, data collection
and analysis, decision to publish, or preparation of
the manuscript. The specific roles of these authors
are articulated in the 'author contributions' section.

**Competing interests:** The authors have read the
journal's policy and have the following competing
interests: author LM received fees for a medical
speech about the results of the study at an AMGEN
conference. This does not alter our adherence to
PLOS ONE policies on sharing data and materials.
There are no patents, products in development or
marketed products associated with this research to
declare.

## Conclusion

No influence of the dialysate calcium concentration was evidenced on survival whereas all MBD therapies were associated with a survival improvement depending on the daily dose used.

## Introduction

International guidelines don't recommend the use of a calcium concentration above 1.5 mmol/L that provides a per-dialysis calcium load and is associated with the progression of vascular calcification [1]. This worsening was correlated with the calcium load [2]. Similarly, the oral calcium supplementation is related to vascular calcifications progression and the switch to non-calcium phosphate-binders is followed by an improvement in their progression [3, 4, 5]. The 2003 Dialysis Outcomes Quality Initiative advised a 1.25 mmol/L dialysate calcium concentration and an oral calcium load of below 2 g including food intake [6]. The European Renal Association–European Dialysis and Transplant Association (ERA-EDTA) guidelines recommend a personalized dialysate calcium concentration [7]. Lastly, the Kidney Disease Improving Global Outcomes (KDIGO) 2009 guidelines advised a dialysate calcium concentration varying from 1.25 to 1.5 mmol/L but the recommendation was graded 2D [8]. The KDIGO 2017 guidelines give little advice on this topic, and simply state that 1.25 mmol/L is the calcium concentration that allows a neutral calcium balance [9].

In the Dialysis Outcome Practice Patterns Study (DOPPS) published in 2008, a 1.25 mmol/L dialysate calcium concentration was used in less than 5% of centers in France, and the 1.75 mmol/L was still frequently used [10]. We conducted a nationwide observational, longitudinal study on the use of dialysate calcium concentrations and of the mineral bone disease (MBD) treatment over the 2010–2014 period in France and their relations with survival.

## Methods

### Population

Data are extracted from REIN registry, which includes all patients with end-stage kidney disease in France on chronic renal replacement therapy. Details of methods and quality control of the REIN registry have been described elsewhere [11]. Data were fully anonymized. Approvals from the National Commission on Informatics and Liberty and from the Advisory Committee on Information Processing in Material Research in the Field of Health were obtained through the national REIN registry. The patients have an opt out option if they don't want to be included into the registry and the patients associations are participating to the monitoring of the registry. We included all incident adult patients having initiated dialysis between January 1st, 2010, and December 31st, 2013, and who were dialyzed for more than 3 months (flow chart, Fig 1).

### Calcium dialysate exposure

Exposures to the different dialysate calcium concentrations were constructed from the sales data. All the dialysate manufacturers operating in France (Soludia Bellco now Medtronic, Fresenius, Baxter-Gambro, Hemotech, Fresenius Medical Care, and BBraun) provided the number, the calcium concentration and the acid type of dialysate bags sold by year and by dialysis center from 2010 to 2014. The calcium load from citric acid dialysate is lower than the one

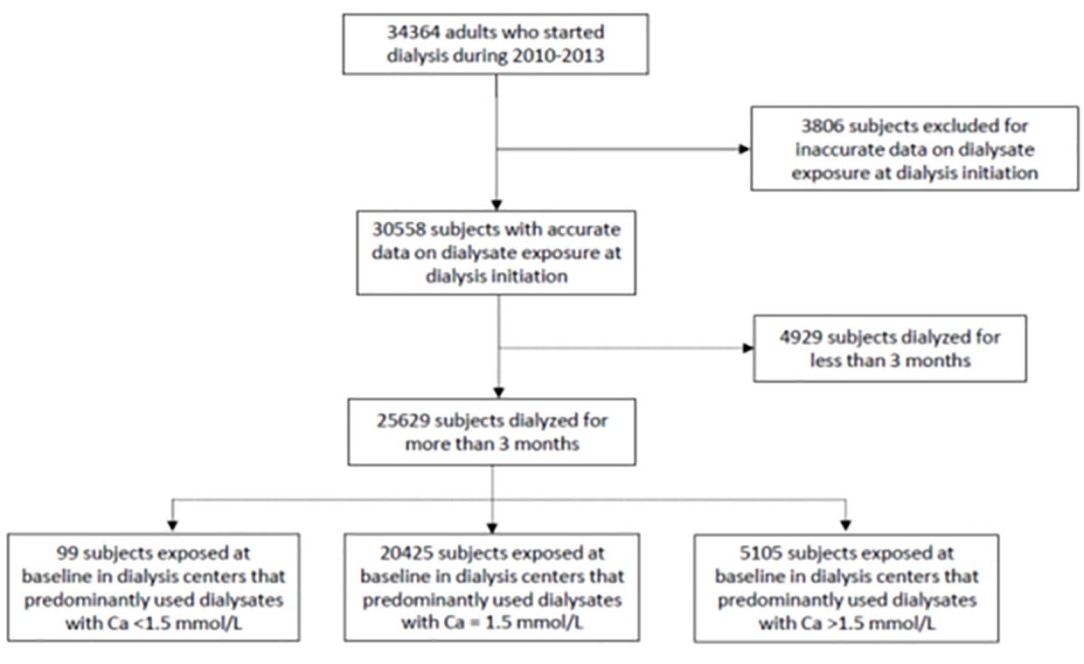

**Fig 1. Flow chart.**

from standard acetic acid dialysate or HCl dialysate; to deliver the same calcium load, the calcium concentration of a citric acid dialysate has to be about 0.15 mmol/l higher than in the other two dialysates. Because the purpose of the study is to analyze the effect of the calcium load on survival, the citric dialysate was systematically reclassified to a -0.15 mmol/L lower calcium concentration. The dialysis centers were classified yearly with regard to their percentage use of standard, citric acid and hydrochloric acid dialysates (100% standard dialysate being the reference).

To ensure the quality of the exposure assessment, we compared the number of patients on hemodialysis estimated from the yearly dialysate volume sold to each unit with the number of patients actually reported each year by the units to the REIN registry. For each dialysis unit, we calculated the ratio of the number of dialysis patients provided by the national registry to the number estimated from the volume of dialysate sold; this ratio defined the percentage of dialysate exposure by center and by year. Dialysate exposure ratios < 1.2 were considered accurate, reflecting that reported sales covered dialysis needs. Patients who started dialysis in a unit with a ratio >1.2 were not included in the study or were censored at the time that dialysate exposure data were considered inaccurate.

Exposure at the unit level was assigned at the patient level by time period according to the yearly changes of dialysate exposure by center and to each patient's changes of dialysis center.

## MBD therapies

Individual data on MBD therapies were extracted from French national health database which retrieved the therapies bought by each subject in number of canister per month, recalculated as a dose per day. This access allows evaluating therapies exposure by day and by drugs dose. Oral calcium, native and active vitamin D (1α calcidol), sevelamer, lanthanum and cinacalcet were studied. On the 25629 subjects included, 21497 patients were identified in the database. In the lack of unique identifier between French national health database and the REIN registry

database, we proceeded to a stepwise indirect linkage using the following data: gender, age, residency code, a national hospital identifier, and date of dialysis start.

## Data

The REIN data at dialysis start included age, gender, nephropathy, emergency start, laboratory results (albumin, hemoglobin, and estimated glomerular filtration rate (eGRF)), and comorbidities. We used the Chronic Kidney Disease Epidemiology Collaboration (CKD-EPI) equation to calculate eGFR. Baseline comorbidities included active malignancy, cirrhosis, handicap status, obesity (body mass index $\geq$30 kg/m$^2$), arrhythmia, respiratory insufficiency, heart failure, and at least one arterial disease (among stroke, transient ischemic attack, coronary insufficiency, abdominal aortic aneurysm, and lower limb arteritis). Furthermore, treatment data including frequency and duration of dialysis, hemodiafiltration use, vascular access and center type (center, medicalized unit or self-assisted unit) were updated annually and when the patient moved to another dialysis center. The events registered included kidney transplantation, transfer to peritoneal dialysis, weaning from dialysis, and death through December 31, 2014.

## Statistical analysis

Patient's baseline characteristics and technical data were compared according to dialysate calcium exposure at dialysis start in 2 groups: mainly exposed to a calcium concentration $\leq$1.5 mmol/L or >1.5 mmol/l. MBD therapies are shown at baseline in these 2 groups and on the subgroups of 8179 subjects followed for at least 3 years. Categorical and continuous covariates at inclusion were compared between groups using Fisher exact test, Pearson Chi-Square or one-way ANOVA as appropriate. Crude events rates were described according to dialysate calcium exposure at baseline in 2 groups $\leq$1.5 mmol/L or >1.5 mmol/L (gathering 1.6 and 1.75 mmol/L).

We used Cox proportional hazard risk models to estimate hazard ratios (HR) and their 95% confidence intervals (95% CI) for all-cause mortality associated with the time-varying dialysate calcium and MBD therapies exposures. The percentage of use per center of dialysate calcium lower and higher than 1.5 mmol/L, and the MBD therapies were analyzed in tertiles, zero use being the reference. The Cox models were stratified by center type. The Cox model was also adjusted for age, gender, co-morbidities, dialysis start in emergency, biological covariates at dialysis start, vascular access type, dialysis session length, hemodiafiltration, number of dialysis sessions per week. These last four variables were also included as time-dependent covariates. Because some unknown patient characteristics and medical practice patterns may vary by unit, robust variance estimates (by a sandwich estimator) were used to account for unit clustering effect [12]. The proportionality hazards assumption was tested by the Schoenfeld residual method. Survival times were censored at the time of event for kidney transplantation, weaning from dialysis, loss to follow-up, moving out of France, transfer to a dialysis unit with inaccurate dialysate exposure, peritoneal dialysis, or home dialysis.

The all-cause and cardiovascular mortality models were lastly adjusted for the dialysate acid type classified yearly for citric acid, HCl and standard as a time dependent covariate.

Missing values on adjustment covariates were treated by multiple imputations using multivariate imputation by chained equations (MICE, packages mice and miceadds of R). Twenty iterations were used and 20 imputed datasets were created. All covariates presented in the Cox models were included in the imputation procedure. Sixty percent of patients had at least one missing data on adjustment covariates. Each Cox model was performed on the 20 imputed datasets and these results were pooled by Rubin's rules.

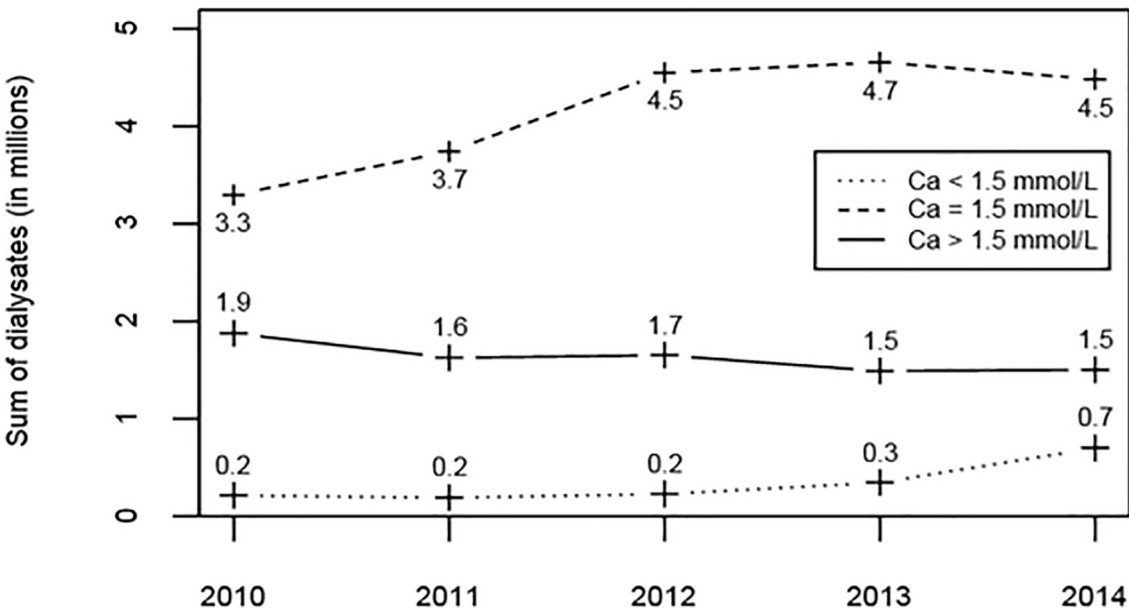

**Fig 2. Overall sales of dialysate in France from 2010 to 2014, as a function of the calcium concentration.**

All tests were two-tailed, and the threshold for statistical significant was set to $p < 0.05$. All statistical analyses were performed with SAS software (version 9.4) and R software.

## Results

The exposure was dominated by the 1.5 mmol/L concentration with 3.3 millions of dialysate bags sold in 2010 and up to 4.5 millions in 2014 (Fig 2). The >1.5 mmol/L concentration decreased from 1.9 million bags to 1.5 million whereas <1.5 mmol/L increased from 0.2 to 0.7 million bags during the same period. In 2010, the <1.50 mmol/L concentration was used at a median level of 2.5% in each center (IQR 0.6%-4.2%) that slightly increased after to 4.4% (IQR 2%-11.7%). The >1.5 mmol/L was used at a median level of 24.5% by center (IQR 11%-45.2%) that decreased thereafter to 15.1% (IQR 6.7%-34.3%, Fig 3).

The 1.5 mmol/L was used in 97 to 99% of the dialysis centers on the study period and was the main dialysate in 81 to 83% of the centers (Table 1). However and applying European guidelines for a personalized prescription, 78 to 86% of the dialysis centers from 2010 to 2014 used concomitantly 3 calcium concentrations, <1.5, 1.5 and >1.5 mmol/L which was the most frequent combination. The second most frequent combination was 1.5 and >1.5 mmol/L, used in 5 to 12% of the centers. The use of a unique calcium concentration remained rare as the combination of 2 concentrations that included <1.5 mmol/L. Still 19% in 2010 to 14% in 2014 of the dialysis centers used >1.5 mmol/L as the main dialysate, percentage that decreased on the time period to the profit of the combination of 3 dialysates that steadily increased. Citric dialysate after requalification of the calcium concentration was the most frequently dialysate used with a concentration <1.5 mmol/L and standard dialysate was the most used with a concentration >1.50 mmol/L (Table 2).

Baseline patients characteristics differed between the 2 groups ≤1.5 mmol/L or >1.5 mmol/L of dialysate calcium for diabetes, arterial disease, initial nephropathy, start in emergency, hemodiafiltration use, vascular access, units types, sessions per week, sessions time, CKD-EPI eGFR and dialysate acid type (Table 3). Native vitamin D was the most frequent MBD therapy prescribed to 56% of the subjects during the first year (Table 4), being less

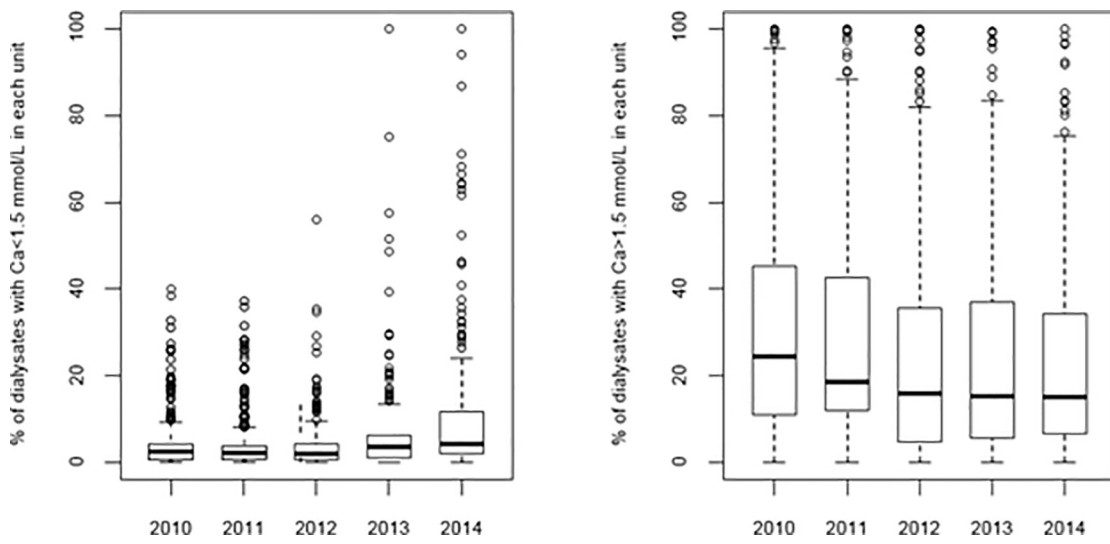

**Fig 3. Boxplots of the dialysis units' percentage use of dialysate calcium concentrations between 2010 and 2014.**

prescribed in the >1.5 mmol/L dialysate group. Active vitamin D was far less prescribed in 15.8% at a median dose of 0.23 µg/d (IQR 0.1–0.35). Oral calcium was prescribed to 42% of the subjects at a median dose of 997 mg/d (IQR 508–1736) during the first year and at a higher dose and more frequently in the >1.50 mmol/L dialysate group. This latter group was therefore exposed to a markedly higher calcium load. Sevelamer was the most frequent non-calcium based phosphate binder used, prescribed to 30.7% of the patients during the first year at a median dose of 2526 mg/d (IQR 1419–3975). Finally, lanthanum was used by 10.9% of the subjects at a median dose of 1098 mg/d (IQR 506–1849) during the first year. The phosphate

**Table 1. Dialysate calcium concentrations by dialysis center in France from 2010 to 2014.**

| N = 1214 | 2010 (N = 1080) | 2011 (N = 1091) | 2012 (N = 1120) | 2013 (N = 1119) | 2014 (N = 1118) |
|---|---|---|---|---|---|
| **Uses (several answers possible)** | | | | | |
| Ca<1.5 mmol/L | 82% (886) | 84% (912) | 84% (942) | 89% (997) | 91% (1016) |
| Ca = 1.5 | 97% (1045) | 97% (1062) | 98% (1103) | 99% (1113) | 98% (1101) |
| Ca>1.5 | 94% (1015) | 93% (1017) | 92% (1028) | 93% (1042) | 92% (1028) |
| **Combinations used** | | | | | |
| Ca<1.5 mmol/L | - | - | - | 0.1% (1) | 0.3% (3) |
| Ca = 1.5 | 4% (41) | 5% (54) | 5% (59) | 4% (50) | 3% (38) |
| Ca>1.5 | 2% (18) | 0.9% (10) | 1% (15) | - | 0.4% (4) |
| Ca<1.5/Ca = 1.5 | 2% (24) | 2% (20) | 3% (33) | 2% (26) | 4% (49) |
| Ca = 1.5/Ca>1.5 | 12% (135) | 11% (115) | 9% (104) | 6% (72) | 5% (60) |
| Ca<1.5/Ca>1.5 | 2% (17) | 2% (19) | 0.2% (2) | 0.4% (5) | 0.9% (10) |
| Ca<1.5/Ca = 1.5/Ca>1.5 | 78% (845) | 80% (873) | 81% (907) | 86% (965) | 85% (954) |
| **Majority use** | | | | | |
| Ca < 1.5 mmol/L | - | - | 0.1% (1) | 0.4% (5) | 4% (42) |
| Ca = 1.5 | 81% (870) | 82% (894) | 83% (926) | 82% (922) | 82% (916) |
| Ca >1.5 | 19% (210) | 18% (197) | 17% (193) | 17% (192) | 14% (160) |

Data are quoted as the percentage of the dialysis units (N = 1214 at baseline, although centers with poorly defined dialysate exposure were excluded as the study went on).

**Table 2. Percentage of dialysate sales in France from 2010 to 2014, as a function of the calcium concentration and acid type.**

|  | Acetate | | | HCl | | | Citrate | | |
|---|---|---|---|---|---|---|---|---|---|
|  | Ca<1.5 | Ca = 1.5 | Ca>1.5 | Ca<1.5 | Ca = 1.5 | Ca>1.5 | Ca<1.5 | Ca = 1.5 | Ca>1.5 |
| 2010 | 4% | 61% | 35% | 3% | 76% | 21% | - | - | - |
| 2011 | 3% | 67% | 30% | 2% | 78% | 19% | 100% | - | - |
| 2012 | 4% | 70% | 27% | 3% | 86% | 12% | 99% | - | 1% |
| 2013 | 4% | 72% | 25% | 2% | 90% | 8% | 76% | 14% | 10% |
| 2014 | 3% | 73% | 23% | 2% | 84% | 14% | 73% | 8% | 19% |

binders were slightly more prescribed in the ≤1.5 mmol/L dialysate calcium group. Cinacalcet was the less frequently MBD therapy prescribed to 8.4% of the subjects, slightly more in the ≤1.5 mmol/L dialysate calcium group and surprisingly at a median dose of 27 mg/d (IQR 14–37) lower than the first pill dosage.

In the subgroup of patients with at least 3 years of follow-up data, the prescription frequency and median dose fell over time for native vitamin D, oral calcium, phosphate binders and active vitamin D (Table 5). For example, the prescription frequency and median [IQR] dose of oral calcium fell from 47% to 34% and from 1068 mg/day [577–1801] to 829 mg/day [411–1493], respectively. The prescription frequency and median [IQR] dose of sevelamer fell from 33.5% to 26.9% and from 2762 mg/day [1578–4267] to 2323 mg/day [1184–3748], respectively. These courses combined the prescription and the adherence trends. Cinacalcet was the only MBD treatment prescribed more frequently during the follow-up period, although the median dose remained abnormally low.

The death and transplantation rates for 100 person-years were quite similar between the groups ≤1.5 mmol/L and the >1.5 mmol/L as the other events (Table 6). In the full adjusted Cox analyses, the dialysate calcium concentrations did not influence survival (Table 7). Using 2 or 3 calcium formulas compared to one brought no survival benefit (HR 1.14 95%CI 0.85–1.52 for 2 formulas, HR 1.04 95%CI 0.78–1.4 for 3 formulas).

A daily dose in the second and third tertiles of calcium, active vitamin D, native vitamin D, sevelamer, lanthanum, cinacalcet were associated with decreased HRs for all-cause mortality (Table 7). Accordingly the cardiovascular mortality HR was decreased by a daily dose of these drugs in the same tertiles. Surprisingly some of the first tertiles of these drugs were associated with a deleterious effect.

## Discussion

First of all, we depicted the landscape of the mineral bone disease therapies at the patient level and the use of the dialysate calcium concentrations at the center level in our country from 2010 to 2014. Dialysate >1.5 mmol/L remained prescribed more than the dialysate <1.5 mmol/L known to avoid a calcium load during the session. The > 1.5 dialysate was however not associated with a worse survival in our study. Nephrologists were trained in programs employing several formulas of dialysate calcium, the combination of 3 ones being common. ERA-EDTA guidelines recommend a personalized prescription of the dialysate calcium. Our full adjusted Cox models did not evidence a relevant survival benefit associated with centers that used more than one dialysate calcium concentration. Lastly, most of the MBD treatments were associated with longer survival.

Our results for the dialysate calcium concentration contrast with recent observational and randomized studies. The DOPP Study evidenced a mortality risk increased by 13% with a high dialysate calcium and a lower risk of parathyroidectomy [13]. One cohort study found a

**Table 3. Characteristics of the population at dialysis initiation, as extracted from the REIN registry after multiple imputations and as a function of the main dialysate calcium concentration used in the baseline unit.**

| N = 25629 | All (N = 25629) | Main facility-level dialysate Ca concentration mmol/L | | p-value | Missing |
|---|---|---|---|---|---|
| | | Ca ≤ 1.5 (N = 20524) | Ca >1.5 (N = 5105) | | |
| Age (years) | 70.4 (59.1–79.4) | 70.3 (59–79.5) | 70.8 (59.7–79.2) | 0.376 | |
| Sex (% males) | 63% (16140) | 63% (12950) | 62% (3190) | 0.429 | |
| Diabetes | 42% (10786) | 42% (8525) | 44% (2261) | <0.001 | 129 |
| Respiratory failure | 14% (3662) | 14% (2922) | 14% (739) | 0.685 | 824 |
| Cirrhosis | 2% (534) | 2% (445) | 2% (89) | 0.069 | 694 |
| Cancer | 11% (2889) | 11% (2324) | 11% (565) | 0.618 | 687 |
| Heart failure | 25% (6319) | 25% (5085) | 24% (1234) | 0.396 | 658 |
| Cardiac rhythm disorder | 21% (5414) | 21% (4330) | 21% (1083) | 0.858 | 708 |
| Peripheral arterial disease | 39% (9962) | 39% (7905) | 40% (2057) | 0.021 | |
| Body mass index ≥ 30 kg/m$^2$ | 23% (5797) | 23% (4622) | 23% (1175) | 0.498 | 5730 |
| Walking | | | | 0.88 | 2545 |
| Normal | 82% (21096) | 82% (16904) | 82% (4192) | | |
| Able with help | 13% (3314) | 13% (2650) | 13% (664) | | |
| Unable with help | 5% (1219) | 5% (971) | 5% (249) | | |
| Nephropathy | | | | <0.001 | |
| Vascular or hypertensive nephropathy | 27% (6896) | 27% (5610) | 25% (1286) | | |
| Diabetic nephropathy | 23% (5966) | 23% (4693) | 25% (1273) | | |
| Glomerulopathies | 10% (2633) | 11% (2165) | 9% (468) | | |
| Polycystic kidney disease | 6% (1526) | 6% (1231) | 6% (295) | | |
| Tubulointerstitial nephropathy | 4% (1082) | 4% (871) | 4% (211) | | |
| Other or unknown diseases | 29% (7526) | 29% (5954) | 31% (1572) | | |
| Emergency dialysis initiation | 32% (8119) | 31% (6431) | 33% (1688) | 0.022 | 1377 |
| Hemodiafiltration | 12% (2986) | 12% (2512) | 9% (474) | <0.001 | |
| Vascular access | | | | <0.001 | 1436 |
| Native fistula or graft | 53% (13466) | 52% (10732) | 54% (2734) | | |
| Catheter | 41% (10527) | 42% (8571) | 38% (1956) | | |
| Other | 6% (1635) | 6% (1221) | 8% (415) | | |
| Unit type | | | | 0.006 | |
| Centre | 91% (23335) | 91% (18746) | 90% (4589) | | |
| Medicalized unit | 3% (800) | 3% (607) | 4% (193) | | |
| Self dialysis unit | 2% (420) | 2% (329) | 2% (91) | | |
| Training | 4% (1074) | 4% (842) | 5% (232) | | |
| Sessions per week | | | | <0.001 | 176 |
| 2 | 6% (1608) | 7% (1379) | 4% (229) | | |
| 3 | 92% (23685) | 92% (18865) | 94% (4820) | | |
| Other | 1% (336) | 1% (280) | 1% (55) | | |
| Dialysis time > 4 hours | 3% (717) | 2% (471) | 5% (246) | <0.001 | 161 |
| Albuminemia (g/l) | | | | 0.601 | 10152 |
| <25 | 11% (2781) | 11% (2217) | 11% (563) | | |
| [25;30[ | 18% (4675) | 18% (3738) | 18% (937) | | |
| [30;35[ | 30% (7638) | 30% (6160) | 29% (1479) | | |
| > = 35 | 41% (10535) | 41% (8409) | 42% (2125) | | |
| Hemoglobin level (g/dl) | 10.1 (9–11.1) | 10.1 (9–11.1) | 10 (9–11.1) | 0.387 | 4837 |

*(Continued)*

**Table 3.** (Continued)

| N = 25629 | All (N = 25629) | Main facility-level dialysate Ca concentration mmol/L | | p-value | Missing |
| | | Ca ≤ 1.5 (N = 20524) | Ca >1.5 (N = 5105) | | |
|---|---|---|---|---|---|
| eGFR> 10 ml/min/1.75m2 | 26% (6766) | 27% (5458) | 26% (1308) | 0.184 | 3863 |

Data are presented as the median [interquartile range (IQR)] or the percentage (n). The p-value was calculated with Fisher's exact test, Pearson's chi-squared test or a one-way analysis of variance, as appropriate. Peripheral arterial disease included stroke, transient ischemic attack, coronary heart failure, aneurysm of the abdominal aorta or arteritis of the lower limbs. CKD-EPI eGFR: the glomerular filtration rate estimated using the Chronic Kidney Disease-Epidemiology Collaboration equation.

cardiovascular mortality risk multiplied by 5.44 (95%CI 2.5–11.7) associated with the use of 1.75 mmol/l dialysate calcium in subjects having a high PTH level at inclusion [14]. One register study on 1182 subjects incident in dialysis associated the use of 1.75 mmol/L dialysate calcium with an all-cause mortality HR of 3.67 (95%CI 1.7–7.5) compared to the 1.25 mmol/L formula and of 2.23 (95%CI 1.2–3.9) compared to the 1.5 mmol/L, results confirmed in a subset of patients matched by a propensity score [15]. To the best of our knowledge, the association between survival and dialysate calcium has not been assessed in recent randomized studies. One trial evaluated the coronary artery calcifications of subjects randomized to 1.25 mmol/L versus 1.75 mmol/L and evidenced the higher progression rate when dialyzed with 1.75 mmol/L, especially in patients with uncontrolled phosphatemia [1]. Our study was unable to evidence any benefit from the use of 1.25 mmol/L formula probably because of its scarce use with a median percentage per unit below 5%. One randomized study having included patients with parathyroid hormone lower than 2 x normal showed a faster progression of the vascular

**Table 4. First-year medication prescriptions as a function of the primary facility-level dialysate calcium concentration at baseline, for the 21497 patients identified in the French national health database.**

| | All (N = 21497) | Main facility-level dialysate Ca concentration | |
| | | Ca≤1.5 (N = 17135) | Ca>1.5 (N = 4362) |
|---|---|---|---|
| **Active vitamin D** | | | |
| % of patients exposed | 15.8% | 15.7% | 16% |
| Dose μg/d | 0.23 (0.1–0.35) | 0.24 (0.1–0.37) | 0.21 (0.08–0.29) |
| **Native vitamin D** | | | |
| % of patients exposed | 56% | 57% | 51.8% |
| Dose UI/d | 1948 (986–3288) | 1953 (989–3288) | 1920 (910–3231) |
| **Calcium** | | | |
| % of patients exposed | 42.8% | 41.9% | 46.4% |
| Dose mg/d | 997 (508–1736) | 986 (506–1721) | 1035 (530–1779) |
| **Cinacalcet** | | | |
| % of patients exposed | 8.4% | 8.5% | 7.9% |
| Dose mg/d | 27 (14–37) | 27 (15–38) | 25 (11–35) |
| **Lanthanum** | | | |
| % of patients exposed | 10.9% | 11% | 10.4% |
| Dose mg/d | 1098 (506–1849) | 1085 (500–1878) | 1125 (536–1737) |
| **Sevelamer** | | | |
| % of patients exposed | 30.7% | 30.8% | 29.9% |
| Dose mg/d | 2526 (1419–3975) | 2526 (1412–4012) | 2549 (1426–3945) |

data are presented as (i) the percentage of patients having received the drug, and (ii) the median [interquartile range] daily dose among patients having received the drug.

**Table 5. Medications prescriptions as a function of the main facility-level dialysate calcium concentration, for the 8179 patients with at least 3 years of follow-up and having been identified in the French national health database.**

| | Year of follow-up | | |
|---|---|---|---|
| **Drug x Main dialysate used** | **1** | **2** | **3** |
| **Ca ≤ 1.5 mmol/L** | N = 6332 | N = 6586 | N = 6703 |
| **Ca > 1.5 mmol/L** | N = 1847 | N = 1593 | N = 1476 |
| **Active vitamin D (µg/d)** | % Median **µg/d** (IQR) | % Median **µg/d** (IQR) | % Median **µg/d** (IQR) |
| Total | 19.4% 0.24 (0.1–0.37) | 14% 0.25 (0.12–0.37) | 11.7% 0.23 (0.12–0.35) |
| Ca ≤ 1.5 mmol/L | 19.2% 0.25 (0.11–0.39) | 14% 0.25 (0.12–0.37) | 11.8% 0.23 (0.12–0.35) |
| Ca > 1.5 mmol/L | 20.1% 0.22 (0.08–0.3) | 14.1% 0.21 (0.1–0.29) | 11.6% 0.23 (0.14–0.35) |
| **Native vitamin D (UI/d)** | % Median (IQR) | % Median (IQR) | % Median (IQR) |
| Total | 59% 2067 **UI/d** (1096–3288) | 57.1% 1918 **UI/d** (822–3014) | 50.4% 1644 **UI/d** (822–2904) |
| Ca ≤ 1.5 mmol/L | 60.3% 2077 **UI/d** (1111–3289) | 58.1% 1918 **UI/d** (826–3014) | 51.4% 1644 **UI/d** (822–2879) |
| Ca > 1.5 mmol/L | 54.5% 2046 **UI/d** (907–3288) | 53% 1808 **UI/d** (822–2999) | 46.2% 1668 **UI/d** (822–3014) |
| **Calcium (mg/d)** | % Median (IQR) | % Median (IQR) | % Median (IQR) |
| Total | 47.4% 1068 **mg/d** (577–1801) | 40.4% 928 **mg/d** (493–1578) | 34.3% 829 **mg/d** (411–1493) |
| Ca ≤ 1.5 mmol/L | 45.9% 1068 **mg/d** (575–1777) | 39.9% 928 **mg/d** (493–1578) | 33.8% 835 **mg/d** (411–1493) |
| Ca > 1.5 mmol/L | 52.3% 1046 **mg/d** (594–1825) | 42.9% 904 **mg/d** (493–1529) | 36.5% 822 **mg/d** (411–1519) |
| **Cinacalcet (mg/d)** | % Median (IQR) | % Median (IQR) | % Median (IQR) |
| Total | 9.4% 28 **mg/d** (15–37) | 12.8% 25 **mg/d** (14–38) | 13.2% 25 **mg/d** (14–35) |
| Ca ≤ 1.5 mmol/L | 9.6% 28 **mg/d** (16–40) | 12.8% 28 **mg/d** (14–39) | 13.3% 25 **mg/d** (14–34) |
| Ca > 1.5 mmol/L | 8.8% 25 **mg/d** (14–34) | 12.7% 23 **mg/d** (12–35) | 12.9% 26 **mg/d** (12–39) |
| **Lanthanum (mg/d)** | % Median (IQR) | % Median (IQR) | % Median (IQR) |
| Total | 12.4% 1171 **mg/d** (554–1849) | 12.1% 992 **mg/d** (493–1807) | 10.3% 952 **mg/d** (493–1726) |
| Ca ≤ 1.5 mmol/L | 12.6% 1154 **mg/d** (534–1911) | 12.2% 998 **mg/d** (493–1826) | 10.5% 925 **mg/d** (493–1726) |
| Ca > 1.5 mmol/L | 11.9% 1220 **mg/d** (608–1788) | 11.4% 986 **mg/d** (493–1792) | 9.2% 1163 **mg/d** (691–1726) |
| **Sevelamer (mg/d)** | % Median (IQR) | % Median (IQR) | % Median (IQR) |
| Total | 33.5% 2762 **mg/d** (1578–4267) | 31.4% 2367 **mg/d** (1184–3945) | 26.9% 2323 **mg/d** (1184–3748) |
| Ca ≤ 1.5 mmol/L | 34% 2762 **mg/d** (1582–4299) | 31.2% 2367 **mg/d** (1184–3945) | 26.6% 2348 **mg/d** (1184–3854) |
| Ca > 1.5 mmol/L | 31.8% 2618 **mg/d** (1554–4142) | 32.1% 2361 **mg/d** (1184–3551) | 28.3% 2170 **mg/d** (1184–3551) |

data are presented as (i) the percentage of patients having received the drug, and (ii) the median [interquartile range] daily dose among patients having received the drug.

calcifications in subjects dialyzed with 1.5 mmol/l instead of 1.25 [16]. Lastly, a North-American study analyzed dialysis centers using a very low dialysate calcium concentration (1.0 mmol/L) and defined a facility level covariate [17]. They found that those using 1.25 mmol/L in less than 75% of the subjects, the remaining being dialyzed on 1.0 mmol/L had a similar mortality risk than those using 1.25 mmol/L in more than 75% of the subjects. However, hospitalizations for cardiac failure, hypotension, hypocalcemia, and the use of MBD treatments were more frequent with the 1.00 mmol/l formula.

Similarly to the low dialysate calcium, the calcium free phosphate binders were shown to slow the vascular calcifications rate [5, 18, 19]. This effect might be a useful surrogate endpoint for survival. In a recent meta-analysis, sevelamer lowered mortality compared to calcium [20]. We have however no proof of the superiority of sevelamer over placebo because the initial trials did not feature a placebo arm. Therefore we cannot be totally confident that these trials evidenced the harmful effect of calcium or/and the benefit from sevelamer. In the present observational study, both treatments appeared to be beneficial on survival. Probably the calcium based phosphate-binder benefit should also be viewed in the perspective of a median

**Table 6. Person-years of exposure and percentages of events, by dialysate group at baseline.**

| N = 25629 | Main facility-level calcium concentration at baseline | |
| --- | --- | --- |
| | Ca ≤ 1.5 mmol/L (N = 20524) | Ca >1.5mmol/L (N = 5105) |
| Person-years of exposure | 36667.4 | 9764.1 |
| **Events% (n)** | | |
| At home | 0.2% (51) | 0.2% (8) |
| Deceased | 27% (5495) | 28% (1451) |
| Loss to follow-up | 0.4% (84) | 0.4% (19) |
| Moved out to France | 0% (6) | 0.1% (7) |
| Switched to peritoneal dialysis | 0.9% (188) | 0.7% (35) |
| Switched to a censored dialysis center | 5% (1045) | 4% (227) |
| Transplanted | 12% (2475) | 12% (598) |
| Discontinuation of dialysis | 4% (767) | 4% (208) |
| Total events | 49% (10111) | 50% (2553) |
| **Rate for 100 person-years** | | |
| Mortality rate | 15% | 14.9% |
| Transplantation rate | 6.7% | 6.1% |

dose around 1 g/d that remained very much lower than the dose prescribed in the past and that was evidenced to be deleterious [3, 4]. For most of the studied drugs, the first tertile was rather associated with a negative effect on survival that has to be interpreted cautiously. That might have been due to unnecessary prescriptions with more side effects than benefits or poor adherence. Conversely, the large benefit observed with the second and third tertiles for MBD treatments might be due in part to better adherence. Other dietary rules, salt and water restrictions might have been followed more closely by patients with better adherence. They might also have been taking other medications (e.g. antihypertensive and/or antiplatelet drugs) more scrupulously. Hence, selection bias via adherence cannot be ruled out. Furthermore, the group of patients not treated with phosphate binders might have included individuals with low phosphate levels linked to malnutrition. Cinacalcet was markedly taken at a low daily dose, which strongly suggests poor adherence. The EVOLVE trial evidenced a survival benefit from cinacalcet after adjustment for age [21]. Data for etelcalcetide are eagerly awaited. Our present results confirmed this advantage, since the cinacalcet was the MBD treatments associated with the greatest observed benefit. This was the only drug with an increasing frequency over the study period among the dialysis patients followed on at least 3 years. Hyperparathyroidism becomes more frequent with longer dialysis vintage. This worsening might be linked at least in part to the observed decreases in the doses of other drugs taken by these patients. Patients should be made more aware of the benefits and risks of MBD treatments.

Our study has the limitation of an observational one and despite its high power could not overcome some bias. The dialysate calcium concentration was treated as a facility-level covariate, whereas the MBD treatment was treated as a patient-level covariate; these two variables did not suffer from the same biases. As mentioned above, the effect of MBD treatments might have been biased by selection of patients with good adherence for those drugs but probably also for other drugs and for dietary measures. The dialysate covariate could not be influenced by a selection or an indication bias because it was built at the facility level. It might have been biased by a center effect, although this is less probable in view of all the combinations of dialysates used. Even those centers using 3 formulas and being highly involved in applying a personalized prescription were not associated with any benefit. The facility-level covariates precluded any estimates of the time of exposure of each patient to the different dialysate

**Table 7. Fully adjusted hazard ratio (HR) for all-cause and cardiovascular mortality, as a function of exposure to dialysate and to MBD treatments.**

|  | Person-years | All-cause mortality |
|---|---|---|
| % of dialysate with Ca<1.5 mmol/L[1] (ref: 0) | 7654 |  |
| ]0%;2%] | 12168 | 1[0.9;1.12] |
| ]2%;6%] | 15094 | 1.01 [0.9–1.14] |
| >6% | 11516 | 1.05 [0.94;1.18] |
| % of dialysate with Ca>1.5 mmol/L[1] (ref: 0) | 4337 |  |
| ]0;10%] | 13571 | 0.95 [0.84;1.07] |
| ]10%;30%] | 13805 | 0.96 [0.85;1.1] |
| >30% | 14718 | 0.93 [0.82;1.06] |
| Calcium (mg/d, ref: 0)[1] | 32110 |  |
| ]0; 600[ | 4750 | 1.21 [1.11;1.32]* |
| [600; 1200[ | 4080 | 0.77 [0.69;0.86]* |
| ≥ 1200 | 5491 | 0.35 [0.31;0.4]* |
| Active vitamin D (µg/d, ref: 0)[1] | 41472 |  |
| ]0; 0,15[ | 1736 | 1.35 [1.2;1.51]* |
| [0,15; 0,3[ | 1827 | 0.77 [0.64;0.92]* |
| ≥ 0,3 | 1397 | 0.62 [0.46;0.83]* |
| Native vitamin D (UI/d, ref: 0)[1] | 27229 |  |
| ]0; 1200[ | 6661 | 1.15 [1.07;1.25]* |
| [1200; 2400[ | 5256 | 0.65 [0.59;0.71]* |
| ≥ 2400 | 7285 | 0.24 [0.21;0.28]* |
| Sevelamer (mg/d, ref: 0)[1] | 36318 |  |
| ]0; 1600[ | 3463 | 1.34 [1.24;1.46]* |
| [1600; 3200[ | 3344 | 0.62 [0.54;0.7]* |
| ≥ 3200 | 3307 | 0.28 [0.23;0.35]* |
| Lanthanum (mg/d, ref: 0)[1] | 42957 |  |
| ]0; 600[ | 1126 | 1.10 [0.95;1.26] |
| [600; 1200[ | 905 | 0.86 [0.71;1.04] |
| ≥ 1200 | 1444 | 0.31 [0.24;0.41]* |
| Cinacalcet (mg/d, ref: 0)[1] | 43082 |  |
| ]0; 20[ | 1302 | 1.22 [1.05;1.42]* |
| [20; 30[ | 939 | 0.5 [0.37;0.68]* |
| ≥ 30 | 1108 | 0.39 [0.28;0.54]* |
| Missing medication | 7591 | 0.75 [0.7;0.81]* |

Stratified by the center modalities of treatment[1]. Adjusted for hemodiafiltration[1], number of sessions per week[1], vascular access[1], dialysis time > 4 hours[1], sex, respiratory failure, cardiac failure, cirrhosis, cancer, cardiac rhythm disorder, peripheral arterial disease (stroke, transient ischemic attack, coronary failure, aneurysm abdominal aorta or arteritis of the lower limbs), obesity, mobility, initial nephropathy, emergency dialysis initiation, eGFR>10 ml/min/1.75m$^2$, hemoglobin level, albuminemia class.

[1]: time-dependent covariates

*: p<0.05

calcium concentrations. A further study limitation relates to the lack of laboratory data–especially for calcemia, phosphatemia, and the PTH level and on the ultrafiltration rate. The calcium mass balance from the dialysate could not be evaluated. It remains that calcium mass balance is always higher with higher dialysate calcium. We have no way of checking whether the medical teams were applying the guidelines on adjustments as a function of the laboratory

data. The confounding bias induced by the acid type of the dialysate was treated by adjustment. The survivors bias was treated by the time-dependent covariates. The benefit associated with the use of 1.25 mmol/L calcium might have been missed by misclassification and/or by its low rate of use. The combination of the three dialysate calcium concentrations would have been less prone to misclassification because of the stable, frequent levels of use during the study period. Classification of drug use per month was fairly more precise because the data came from a health insurance system that automatically tracks drug purchases for each patient.

In conclusion, our study highlighted the use of various dialysate calcium concentrations across France, and evidenced the centers' widespread use of three dialysate calcium concentrations in line with ERA-EDTA guidelines on personalized prescriptions. The scarce use of the < 1.5 mmol/L calcium dialysate prevented us from drawing firm conclusion on its relation with survival. Lastly, MBD treatments including an adequate supply of calcium were associated with a significant survival advantage; dialyzed patients should be made aware of this advantage by their physician. Cinacalcet was the MBD treatment associated with the greatest survival advantage.

## Acknowledgments

We thank all REIN registry centers, as listed in the REIN network's annual report (http://www.agence-biomedecine.fr/Le-programme-REIN).

## Author Contributions

**Conceptualization:** Lucile Mercadal.

**Formal analysis:** Oriane Lambert, Cécile Couchoud, Marie Metzger.

**Funding acquisition:** Lucile Mercadal.

**Investigation:** Cécile Couchoud, Lucile Mercadal.

**Methodology:** Cécile Couchoud, Gabriel Choukroun, Christian Jacquelinet, Lucile Mercadal.

**Project administration:** Lucile Mercadal.

**Supervision:** Lucile Mercadal.

**Validation:** Gabriel Choukroun, Christian Jacquelinet, Lucile Mercadal.

**Visualization:** Christian Jacquelinet.

**Writing – original draft:** Lucile Mercadal.

**Writing – review & editing:** Lucile Mercadal.

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
