## [Decision Letter · Decision Letter 0]

7 Apr 2020

PONE-D-20-04611

Effects of the dialysate calcium concentrations and mineral bone disease treatments on mortality in The French Renal Epidemiology and Information Network (REIN) registry

PLOS ONE

Dear Dr Mercadal,

Thank you for submitting your manuscript to PLOS ONE. After careful consideration, we feel that it has merit but does not fully meet PLOS ONE’s publication criteria as it currently stands. Therefore, we invite you to submit a revised version of the manuscript that addresses the points raised during the review process.

We would appreciate receiving your revised manuscript by May 22 2020 11:59PM. To enhance the reproducibility of your results, we recommend that if applicable you deposit your laboratory protocols in protocols.io, where a protocol can be assigned its own identifier (DOI) such that it can be cited independently in the future. For instructions see: http://journals.plos.org/plosone/s/submission-guidelines#loc-laboratory-protocols

We look forward to receiving your revised manuscript.

Kind regards,

Tatsuo Shimosawa, M.D., Ph.D.

Academic Editor

PLOS ONE

Journal Requirements:

2. In ethics statement in the manuscript and in the online submission form, please provide additional information about the patient records/samples used in your retrospective study. Specifically, please ensure that you have discussed whether all data/samples were fully anonymized before you accessed them and/or whether the IRB or ethics committee waived the requirement for informed consent. If patients provided informed written consent to have data/samples from their medical records used in research, please include this information.

"LM received fee for a medical speech about the results of the study in an AMGEN conference. "

5. Your ethics statement must appear in the Methods section of your manuscript. If your ethics statement is written in any section besides the Methods, please move it to the Methods section and delete it from any other section. Please also ensure that your ethics statement is included in your manuscript, as the ethics section of your online submission will not be published alongside your manuscript.

Reviewers' comments:

Reviewer's Responses to Questions

**Comments to the Author**

1. Is the manuscript technically sound, and do the data support the conclusions?

Reviewer #1: Yes

Reviewer #2: Yes

Reviewer #3: Yes

Reviewer #4: Yes

2. Has the statistical analysis been performed appropriately and rigorously? 

Reviewer #1: Yes

Reviewer #2: Yes

Reviewer #3: Yes

Reviewer #4: Yes

3. Have the authors made all data underlying the findings in their manuscript fully available?

Reviewer #1: Yes

Reviewer #2: Yes

Reviewer #3: Yes

Reviewer #4: No

4. Is the manuscript presented in an intelligible fashion and written in standard English?

Reviewer #1: Yes

Reviewer #2: Yes

Reviewer #3: Yes

Reviewer #4: Yes

5. Review Comments to the Author

Reviewer #1: The authors demonstrated no significant difference of different calcium concentration of dialysate on the outcome of patients on hemodialysis using REIN registry.

In the analysis, the authors also showed MBD treatment improved the outcome of these patients.

The findings are not always new, but there are some important implications that clinical practice to control MBD is more important than calcium concentration of dialysate using large French sample.

Reviewer #2: In this original paper, Lambert and assocoates analyzed the REIN registry data and reported nn significant difference in mortality depending on dialysate calcium concentration.

Stataical analyses were properly performed.

I have only minor comments.

1. The authors analysed the effects of dialysate calcium concentration mainly, but also analyzed calcium load bythe use of calcium-containing phosphate binders and vitamin D. Thus, the title can be modified to such as 'No effects of calcium load on mortality in French dialysis patients.'

2. The resukts were not consistent with previous papers. The authors should discuss the reasons more in details, especially in terms of French practice pattern, food and life style etc.

Reviewer #3: This is an interesting and well described research from the REIN study group, showing that there is no effect on survival irrespective of dialysate calcium used, this is a little bit in contrast with what is known.

I have some minor comments:

1. it would be of great clinical importance and interest to know how long patients were dialysed against either a DCa of less than 1.5 or above 1.5 or higher and what than the effect on survival might be?

2. do the authors have any information on calcifications of vessels with the different DCa concentrations?

3. Is calcium load or calcium mass balance known?

4. The HR in table 5 suggest that calcium might have a worse effect on survival, could the authors eplain this?

Reviewer #4: Overall: Lambert and colleagues perform a longitudinal analysis of data from the French REIN registry to evaluate the association of facility level DCa use with MBD medication use at the patient level and all-cause mortality after 3 years. They find that the majority of facilities used a DCa concentration of 1.5mmol/L and over the course of 3 years, the proportion of those using >1.5mmol/L decreased while those using <1.5mmol/L increased. The investigators found no effect of higher or lower than DCa 1.5 mmol/L to be associated with mortality and only the MBD therapies were associated with mortality. My comments are below.

Major:

1. Given that the comparison really is looking at DCa <1.5 or DCa >1.5 with the bulk of the population actually using DCa = 1.5, I think the analysis would be better suite being divided into three exposure categories with DCa 1.5 perhaps being the referent group? I worry that any risk of benefit seen the category using DCa <1.5 may be diluted when combined with those using DCa = 1.5. I understand, that there were only relatively few patients with DCa <1.5

2. Can you please explain why the higher DCa group was also exposed to higher dose of oral calcium (Line 168)

3. What is missing from this data is the average volume removal (ultrafiltration) during the dialysis in the various groups. If the reason DCa have an effect of mortality is due to the dose of Ca delivered acutely and chronically, this analysis doesn’t account for Ca losses with ultrafiltration which occur depending on the volume removed- as much as 3mEq of Ca removed with 2L UF. Is there a way to adjust for this? Would the analysis not have been more accurate if the MBD treatments also treated as facility level?

4. Why was this time period chosen? Can you compare data from to 2009 when there may have been a higher prevalence of DCa >1.5mmol/L?

5. The investigators state that their results are in contrast to existing studies, and I agree. However, no clear explanation is provided as to why theirs are either more reliable or a better analytic approach compared to other studies. This discussion is critical and needs to be flushed out better.

Minor:

1. Line 89, should be eGFR

2. The abstract provides no information about the directionality of the association between MBD therapies and mortality

6. PLOS authors have the option to publish the peer review history of their article (what does this mean?). If published, this will include your full peer review and any attached files.

Reviewer #1: Yes: Yoshitaka Ishibashi

Reviewer #2: No

Reviewer #3: No

Reviewer #4: No

---

## [Author Response · Author response to Decision Letter 0]

17 May 2020

Answers to the reviewers 

Data Availability Statement: All data used for this research were extracted from the REIN registry, coordinated and supported by the French Biomedecine Agency. The access to national data is regulated by a scientific committee of French Biomedecine Agency which analyzes each request, and so cannot be made publicly available due to legal restrictions. Data are available upon request. If readers need information about the data from the REIN registry, they can contact Dr. Cecile Couchoud who coordinates the REIN at the national level (email address: cecile.couchoud@biomedecine.fr).

Ethics statement was added line 45 as followed: Approvals from the National Commission on Informatics and Liberty and from the Advisory Committee on Information Processing in Material Research in the Field of Health were obtained through the national REIN registry. The patients have an opt out option if they don't want to be included into the registry and the patients associations are participating to the monitoring of the registry.

Reviewer #1: The authors demonstrated no significant difference of different calcium concentration of dialysate on the outcome of patients on hemodialysis using REIN registry.

In the analysis, the authors also showed MBD treatment improved the outcome of these patients.

The findings are not always new, but there are some important implications that clinical practice to control MBD is more important than calcium concentration of dialysate using large French sample.

Reviewer #2: In this original paper, Lambert and assocoates analyzed the REIN registry data and reported nn significant difference in mortality depending on dialysate calcium concentration.

Stataical analyses were properly performed.

I have only minor comments.

1. The authors analysed the effects of dialysate calcium concentration mainly, but also analyzed calcium load bythe use of calcium-containing phosphate binders and vitamin D. Thus, the title can be modified to such as 'No effects of calcium load on mortality in French dialysis patients.'

Answer: we think that the article is not dealing only with calcium but also with the other treatments for mineral bone disease. So the proposed title is too restricted. In addition, the calcium load of this period is probably far less than before. The decrease of the oral calcium was evidenced by the data of the SNDS and probably made possible that the calcium load of the studied period was no more associated with an increased mortality. This is also true for the dialysate calcium concentration as the calcium concentration > 1.5 decreased over time. It could be: “No effect of reasonable and appropriate calcium load on mortality in the French dialysis patients”, but we did not focus only on calcium. 

2. The resukts were not consistent with previous papers. The authors should discuss the reasons more in details, especially in terms of French practice pattern, food and life style etc.

Answer: No recent randomized study has studied the effect of dialysate calcium load on mortality but only observational ones were published. The sole randomized study disclosed more vascular calcifications with dialysate Ca 1.75 versus 1.25 mmol/L but did not compare Ca 1.5 versus 1.75 or 1.25mmol/L. Additionally they evidenced an effect of dialysate Calcium only in patients having an uncontrolled serum phosphate. At last, the results about dialysate calcium 1.25 mmol/L in our study is limited by the low level of use of this formula. This limitation is discussed as followed : " Our study was unable to evidence any benefit from the use of 1.25 mmol/L formula probably because of its scarce use with a median percentage per unit below 5%." and in the limitation paragraph " The benefit associated with the use of 1.25 mmol/L calcium might have been missed by misclassification and/or by its low rate of use."

Reviewer #3: This is an interesting and well described research from the REIN study group, showing that there is no effect on survival irrespective of dialysate calcium used, this is a little bit in contrast with what is known.

I have some minor comments:

1. it would be of great clinical importance and interest to know how long patients were dialysed against either a DCa of less than 1.5 or above 1.5 or higher and what than the effect on survival might be?

Answer: the time of exposure should be a covariate of interest. But the dialysate calcium concentration covariates were facility-level covariates and not individual-level ones. So for each patient, we only had a probability of exposure to each calcium concentration. From these data, we cannot calculate a time of exposure for each dialysate calcium concentration. We added this comment into the limitation section as followed:

Text: The facility-level covariates precluded any estimates of the time of exposure of each patient to the different dialysate calcium concentrations. 

2. do the authors have any information on calcifications of vessels with the different DCa concentrations?

Answer: we haven’t any data on calcifications in the registry but we discussed this surrogate endpoint in the discussion. 

3. Is calcium load or calcium mass balance known?

Answer: the oral calcium load was estimated from the SNDS data system and was summarized in the treatments tables ( Tables 4 and 5). The calcium load from the dialysate depends on the free calcium concentration of each patient and as discussed into the limitation section, serum calcium is not available in the database of the REIN registry. We added that the calcium mass balance was unknown because of the lack of biology data on serum free calcium in the registry.

Text: A further study limitation relates to the lack of laboratory data – especially for calcemia, phosphatemia, and the PTH level. The calcium mass balance during dialysis sessions could not be evaluated . It remains that calcium mass balance is always higher with higher dialysate calcium.

4. The HR in table 5 suggest that calcium might have a worse effect on survival, could the authors eplain this?

Answer: It might be table 7 as there is no HR in table 5. The dialysate calcium covariate wasn't significantly associated with survival and oral calcium HRs depended on the calcium dose. A negative effect on survival was only significant for the first tertile of dose and was discussed line 303 to 305 and 324 to 326.

Reviewer #4: Overall: Lambert and colleagues perform a longitudinal analysis of data from the French REIN registry to evaluate the association of facility level DCa use with MBD medication use at the patient level and all-cause mortality after 3 years. They find that the majority of facilities used a DCa concentration of 1.5mmol/L and over the course of 3 years, the proportion of those using >1.5mmol/L decreased while those using <1.5mmol/L increased. The investigators found no effect of higher or lower than DCa 1.5 mmol/L to be associated with mortality and only the MBD therapies were associated with mortality. My comments are below.

Major:

1. Given that the comparison really is looking at DCa <1.5 or DCa >1.5 with the bulk of the population actually using DCa = 1.5, I think the analysis would be better suite being divided into three exposure categories with DCa 1.5 perhaps being the referent group? I worry that any risk of benefit seen the category using DCa <1.5 may be diluted when combined with those using DCa = 1.5. I understand, that there were only relatively few patients with DCa <1.5

Answer: In fact, the < 1.5 and 1.5 mmol/L dialysate calcium concentrations exposures are mixed in the patients’ characteristics table (table 3), in the oral treatment description tables (tables 4 and 5) and in the crude events table (table 6) because of its very low level of use. But in the Cox models, the < 1.5, > 1.5 were considered separately. We presented the two covariates DCa < 1.5 and > 1.5, the third DCa 1.5 being in the reference defined by the absence of use of DCa < 1.5 or > 1.5 in table 7. In the DCa > 1.5 covariate, we compared the centers not using this concentration (reference DCa > 1.5 equal to zero) with the centers using it at different percentages of use. In the Dca < 1.5 covariate, we compared the centers not using this concentration (reference DCa < 1.5 equal to zero) with the centers using it at different percentages of use. Therefore in the Cox model, the DCa < 1.5 effect was never diluted into the DCa 1.5 but actually compared to centers only using DCa 1.5 and > 1.5. However we agree that the level of use of DCa < 1.5 was very low and that's why its effect may have been blunted. 

2. Can you please explain why the higher DCa group was also exposed to higher dose of oral calcium (Line 168)

Answer: There is probably a remaining policy in some centers that continue to follow a calcium preferred treatment for hyperparathyroidism and phosphate binding. We found these centers still using calcium as the preferred agent for MBD therapy. But they largely decreased the oral calcium dose and now are more using several formulas instead of Ca > 1.5 alone.

3. What is missing from this data is the average volume removal (ultrafiltration) during the dialysis in the various groups. If the reason DCa have an effect of mortality is due to the dose of Ca delivered acutely and chronically, this analysis doesn’t account for Ca losses with ultrafiltration which occur depending on the volume removed- as much as 3mEq of Ca removed with 2L UF. Is there a way to adjust for this? Would the analysis not have been more accurate if the MBD treatments also treated as facility level?

Answer: indeed we hadn’t any data on ultrafiltration. We add this in the limitation chapter as followed:

Text : A further study limitation relates to the lack of laboratory data – especially for calcemia, phosphatemia, and the PTH level and on the ultrafiltration rate.

Answer: Patient-level and facility-level covariates are exposed to different bias. We cannot ascertain that a facility level variable would have been more accurate than a patient-level covariate. We agree that they are usually complementary. 

4. Why was this time period chosen? Can you compare data from to 2009 when there may have been a higher prevalence of DCa >1.5mmol/L?

Answer: the funding was obtained in 2015 and the analyses built on the 4 previous years to obtain the data of the registry. The DCa > 1.5 mmol/L was still high in France at that period compared to other countries as evidenced in DOPPS and as discussed line 36 (reference 10).

5. The investigators state that their results are in contrast to existing studies, and I agree. However, no clear explanation is provided as to why theirs are either more reliable or a better analytic approach compared to other studies. This discussion is critical and needs to be flushed out better.

Answer: No randomized study was conducted to evaluate the mortality risk with the different dialysate calcium concentrations (line 280). Only one study having evaluated a surrogate endpoint on vascular calcifications is published (line 281-284, reference 1). All other publications are observational and so exposed to similar bias as our study. Our study is however one of those that benefitted from the more exhaustive information with both dialysate calcium concentrations and oral MBD therapies in a large dataset of patients. Randomized studies remain the gold standard for evidence based medicine but are still very expensive and sometimes inconclusive as discussed for EVOLVE (line 320). 

Minor:

1. Line 89, should be eGFR

Answer: the change was made

2. The abstract provides no information about the directionality of the association between MBD therapies and mortality

 Answer: We mentioned that the association depended on the calcium dose. We added that the improvement on survival was significant for the second and third tertiles of dose as followed: 

Text: Depending on the daily dose used, the MBD therapies were associated with survival improvement for calcium, native vitamin D, active vitamin D, sevelamer, lanthanum and cinacalcet in the second and third tertiles of dose.

---

## [Decision Letter · Decision Letter 1]

10 Jun 2020

Effects of the dialysate calcium concentrations and mineral bone disease treatments on mortality in The French Renal Epidemiology and Information Network (REIN) registry

PONE-D-20-04611R1

Dear Dr. Mercadal,

We’re pleased to inform you that your manuscript has been judged scientifically suitable for publication and will be formally accepted for publication once it meets all outstanding technical requirements.

Kind regards,

Tatsuo Shimosawa, M.D., Ph.D.

Academic Editor

PLOS ONE

Additional Editor Comments (optional):

Reviewers' comments:

Reviewer's Responses to Questions

**Comments to the Author**

1. If the authors have adequately addressed your comments raised in a previous round of review and you feel that this manuscript is now acceptable for publication, you may indicate that here to bypass the “Comments to the Author” section, enter your conflict of interest statement in the “Confidential to Editor” section, and submit your "Accept" recommendation.

Reviewer #1: All comments have been addressed

Reviewer #2: All comments have been addressed

Reviewer #3: All comments have been addressed

Reviewer #4: All comments have been addressed

2. Is the manuscript technically sound, and do the data support the conclusions?

Reviewer #1: Yes

Reviewer #2: Yes

Reviewer #3: Yes

Reviewer #4: Yes

3. Has the statistical analysis been performed appropriately and rigorously? 

Reviewer #1: Yes

Reviewer #2: Yes

Reviewer #3: Yes

Reviewer #4: Yes

4. Have the authors made all data underlying the findings in their manuscript fully available?

Reviewer #1: Yes

Reviewer #2: Yes

Reviewer #3: Yes

Reviewer #4: Yes

5. Is the manuscript presented in an intelligible fashion and written in standard English?

Reviewer #1: Yes

Reviewer #2: Yes

Reviewer #3: Yes

Reviewer #4: Yes

6. Review Comments to the Author

Reviewer #1: The authors demonstated the importance of MBD practice rather than calcium concentartion using large French cohort. The result drawn from the study was clinically relevant. I would suggest to accept the paper.

Reviewer #2: The authors responded to my comments and modified the title and modified the limitations as appropriate.

Reviewer #3: (No Response)

Reviewer #4: The authors have addressed my comments appropriately. I do not have any further critiques

Additional discussions points/limitations have been added

7. PLOS authors have the option to publish the peer review history of their article (what does this mean?). If published, this will include your full peer review and any attached files.

Reviewer #1: Yes: Yoshitaka Ishibashi

Reviewer #2: No

Reviewer #3: No

Reviewer #4: No

---

## [Editor Report · Acceptance letter]

17 Jun 2020

PONE-D-20-04611R1 

Effects of the dialysate calcium concentrations and mineral bone disease treatments on mortality in The French Renal Epidemiology and Information Network (REIN) registry 

Dear Dr. Mercadal:

I'm pleased to inform you that your manuscript has been deemed suitable for publication in PLOS ONE. Congratulations! Your manuscript is now with our production department. 

Kind regards, 

on behalf of

Prof. Tatsuo Shimosawa 

Academic Editor

PLOS ONE